# Immune Response to SARS-CoV-2 Vaccine and Following Breakthrough Omicron Infection in an Autoimmune Patient with Hashimoto’s Thyroiditis, Pernicious Anemia, and Chronic Atrophic Autoimmune Gastritis: A Case Report

**DOI:** 10.3390/vaccines10030450

**Published:** 2022-03-15

**Authors:** Emily Cluff, Lorenza Bellusci, Hana Golding, Surender Khurana

**Affiliations:** Center for Biologics Evaluation and Research (CBER), Division of Viral Products, Food and Drug Administration (FDA), Silver Spring, MD 20993, USA; emily.cluff@fda.hhs.gov (E.C.); lorenza.bellusci@fda.hhs.gov (L.B.); hana.golding@fda.hhs.gov (H.G.)

**Keywords:** COVID-19, Omicron, SARS-CoV-2, neutralization, autoimmune, Hashimoto’s, vaccine, mRNA, breakthrough

## Abstract

In healthy adults, hybrid immunity induced by prior SARS-CoV-2 infection followed by two doses of mRNA vaccination provide protection against symptomatic SARS-CoV-2 infection. However, the role of hybrid immunity in autoimmune patients against Omicron is not well documented. Here, we report a young autoimmune patient with prior infection and two doses of mRNA-1273 vaccination who was exposed to Omicron and developed a symptomatic disease. Prior to Omicron infection, the patient had strong neutralizing antibody titers against the vaccine strain, but no neutralization of Omicron. Post Omicron infection, high neutralizing titers against Omicron were observed. Furthermore, enhanced neutralizing antibody titers against other variants of concern—Alpha, Beta, Gamma, and Delta—were observed, suggesting an expansion of cross-reactive memory B-cell response by the SARS-CoV-2 Omicron infection. Autoimmune patients may require careful monitoring of immune function over time to optimize booster vaccine administration.

## 1. Introduction

Since its first appearance in South Africa in November 2021, the SARS-CoV-2 Omicron variant has spread rapidly around the globe. Omicron contains large number of mutations in the SARS-CoV-2 spike protein [1,2,3]. Some of these mutations are known to increase resistance to antibodies elicited by prior infections or current SARS-CoV-2 vaccines [2,4]. The Omicron variant is resistant to most of the MAbs, including those approved for treatment of SARS-CoV-2 exposed individuals [2,4].

For healthy adults, it was shown that hybrid immunity induced by a prior COVID infection followed by two doses of mRNA vaccination, as well as a third vaccine dose, provide efficient protection against symptomatic SARS-CoV-2 disease caused by multiple variants of concern (VOCs) including Omicron [5,6]. However, the durability and breadth of such protection may vary widely among people with comorbidities and different immune status, including autoimmune diseases who are at higher risk of SARS-CoV-2 re-infections. The impact of hybrid immunity induced by prior SARS-CoV-2 infection followed by two doses of mRNA vaccination on Omicron breakthrough infection is unknown in autoimmune patients. Moreover, the role of Omicron infection in immunity against other variants of concern (VOCs) will be important in determining the need for future vaccines. Here, we describe a case report of a young individual infected with Omicron despite prior infection and two doses of mRNA-1273 vaccination. We describe the titers of neutralizing antibodies against the ancestral strain and different VOCs prior to and 14 days following symptomatic Omicron infection.

## 2. Materials and Methods

### 2.1. Ethics Statement

This study was approved by the U.S. Food and Drug Administration’s (FDA) Research Involving Human Subjects Committee (Silver Spring, MD, USA) (FDA-RIHSC-2020-04-02 (252)). This study complied with all relevant ethical regulations for work with human participants, and informed consent was obtained. Samples were collected from individuals who provided informed consent to participate in the study. The participant consented and is a co-author and helped write the manuscript. All assays performed fell within the permissible usages in the original informed consent.

### 2.2. Neutralization Assay

Plasma was evaluated in a qualified SARS-CoV-2 pseudovirion neutralization assay (PsVNA) using SARS-CoV-2 WA1/2020 strain and the Alpha, Beta, Gamma, and Delta variants and the Omicron variant. SARS-CoV-2 neutralizing activity was measured by PsVNA, which was shown to correlate with PRNT (plaque reduction neutralization test with authentic SARS-CoV-2 virus) in previous studies [7,8,9].

Briefly, human codon-optimized cDNA encoding SARS-CoV-2 spike glycoprotein of the WA1/2020 and variant strains were synthesized by GenScript and cloned into eukaryotic cell expression vector pcDNA 3.1 between the BamHI and XhoI sites. The plasmid vector encoding spike for the Omicron variant was a gift from the Vaccine Research Center, NIAID, NIH. Pseudovirions were produced by co-transfection Lenti-X 293T cells with psPAX2(gag/pol), pTrip-luc lentiviral vector and pcDNA 3.1 SARS-CoV-2-spike-deltaC19, using Lipofectamine 3000. The supernatants were harvested at 48 h post transfection and filtered through 0.45 µm membranes and titrated using 293T-ACE2-TMPRSS2 cells (HEK 293T cells that express ACE2 and TMPRSS2 proteins) [9]. Plasma was evaluated in a qualified SARS-CoV-2 pseudovirion neutralization assay (PsVNA) using SARS-CoV-2 WA1/2020 strain and the five variants of concern (VOCs): Alpha variant (B.1.1.7; with spike mutations H69-V70del, Y144del, N501Y, A570D, D614G, P681H, T716I, S982A, and D1118H), Beta variant (B.1.351; with spike mutations L18F, D80A, D215G, L242-244del, R246I, K417N, E484K, N501Y, D614G, and A701V), Gamma variant (P.1; with spike mutations L18F, T20N, P26S, D138Y, R190S, K417T, E484K, N501Y, H655Y, T1027I, D614G, V1176F), Delta variant (B.1.617.2; with spike mutations T19R, G142D, E156del, F157del, R158G, L452R, T478K, D614G, P681R, D950N) and Omicron variant (B.1.1.529; with spike mutations A67V, H69-70del, T95I, G142D, V143-145del, Y145D, N211del, L212I, ins214EPE, G339D, S371L, S373P, S375F, K417N, N440K, G446S, S477N, T478K, E484A, Q493R, G496S, Q498R, N501Y, Y505H, T547K, D614G, H655Y, N679K, P681H, N764K, D796Y, N856K, Q954H, N969K, L981F). SARS-CoV-2 neutralizing activity measured by PsVNA correlates with PRNT (plaque reduction neutralization test with authentic SARS-CoV-2 virus) in previous studies [7,8].

For the neutralization assay, 50 µL of SARS-CoV-2 S pseudovirions (counting ~200,000 relative light units) was pre-incubated with an equal volume of medium containing serial dilutions (20-, 60-, 180-, 540-, 1620-, 4860-, 14,580- and 43,740-fold dilution at the final concentration) of heat-inactivated plasma at room temperature for 1 h. Then, 50 µL of virus–antibody mixture was added to 293T-ACE2-TMPRSS2 cells (10^4^ cells/50 μL), 333 in a 96-well plate. The input virus with all SARS-CoV-2 strains used in the current study was the same (2 × 10^5^ relative light units/50 µL/well). After a 3 h incubation, fresh medium was added to the wells. Cells were lysed 24 h later, and luciferase activity was measured using a One-Glo luciferase assay system (Promega, Madison, WI, USA, Cat# E6130). The assay of each plasma was performed in duplicate, and the 50% neutralization titer was calculated using Prism 9 (GraphPad Software, San Diego, CA, USA). Controls included cells only, virus without any antibody and positive plasma. The limit of detection for the neutralization assay is 1:20. Two independent biological replicate experiments were performed for each sample and variation in PsVNA50 titers was <10% between replicates.

## 3. Results

### Detailed Case Description

Here, we report on a 21-year-old female who was first infected with SARS-CoV-2 in November 2020 when the Alpha variant was predominant (Figure 1A). Patient had severe symptoms for 6 days but was not hospitalized. Post-recovery (2.5 months), she received two doses of Moderna’s mRNA-1273 vaccine on 6 February 2021 and 6 March 2021, respectively. She reported a moderate reaction after the first vaccine, and little to no reaction following the second vaccination.

Between August and September 2021, she was diagnosed with several autoimmune disorders: Hashimoto’s thyroiditis, pernicious anemia, and chronic atrophic autoimmune gastritis, resulting in hypothyroidism and iron deficiency (Figure 1A). She was placed on replacement therapies (25 microgram levothyroxine and 20 milligram duloxetine), but no immunosuppressive drugs.

The patient was exposed again to SARS-CoV-2 on 27 December 2021, with symptoms starting on 28 December 2021, and tested PCR-positive for SARS-CoV-2 Omicron on 1 January 2022. Her symptoms included fever, body aches, fatigue, loss of appetite, cough, difficulty breathing, headaches, rhinitis, and nausea. The most severe symptoms were resolved after 6 days without hospitalization, but low oxygen levels with lingering symptoms persisted for two weeks.

A blood sample was collected from this individual on 3 November 2021, 8 months post second vaccination and prior to Omicron exposure. A second blood sample was obtained two weeks post Omicron infection (PCR-negative) (Figure 1A). The plasma samples were tested for neutralization of the prototype SARS-CoV-2 WA1/2020 as well as the five VOCs—Alpha, Beta, Gamma, Delta and Omicron—using a pseudovirion neutralization assay (PsVNA), as previously described [8], that correlated with the plaque reduction neutralization test of authentic SARS-CoV-2 virus in earlier studies [7,8] (Figure 1B).

Prior to Omicron infection, the patient had strong neutralizing antibody titers against the vaccine strain WA1/2020 (PsVNA50 = 3575). The neutralizing titer against Alpha variant was reduced by 3.9-fold (PsVNA50 = 909.2). The titers against Beta and Gamma variants were reduced by 8.8- and 6.1-fold (PsVNA50 = 408.7 and 591.8, respectively), and were further reduced (20.2-fold) against the Delta (PsVNA50 = 177) variant (Figure 1B). No neutralization of Omicron variant was observed. Therefore, in spite of hybrid immunity, the breadth of anti-SARS-CoV-2 humoral immunity was low in this autoimmune patient by 8 months post second vaccination.

Fourteen days post Omicron infection, the patient’s plasma exhibited strong neutralization of Omicron (PsVNA50 = 2803), while the titer against the vaccine strain, WA1/2020, was unchanged (PsVNA50 = 3343) compared with pre-Omicron infection. However, the neutralization titers were significantly boosted against other VOCs: Alpha (2.7-fold), Beta (3.2-fold), Gamma (2.6-fold), and Delta (3.3-fold) compared with neutralizing antibodies prior to Omicron infection (Figure 1B).

## 4. Discussion

This case report suggests that hybrid immunity generated by prior SARS-CoV-2 infection and mRNA-1273 vaccination was not sufficient to prevent symptomatic infection with Omicron in an autoimmune patient. Prior reports on Omicron breakthrough infections after vaccination with mRNA vaccines did not mention rate of infections among individuals with hybrid immunity and did not report virus-neutralizing titers prior to Omicron infection [10,11,12]. In this case report, we found low titers against several VOC and no titers against Omicron prior to second infection. Yet, this patient developed a robust neutralizing antibody titer against Omicron and surprisingly strongly boosted cross-reactive neutralizing antibodies against the other VOCs, while the response against the ancestral prototype spike (Wuhan-like) that was in the two vaccine doses was not boosted. The Omicron infection resulted in de novo activation of Omicron-specific B cells as well as recalled cross-reactive memory B cells that had undergone affinity maturation, resulting in a broad neutralizing antibody response against multiple variants [6,13,14]. The limitation of this study is the single case. However, the findings warrant further evaluation of breakthrough infections in individuals with hybrid immunity. Furthermore, the outcome of vaccination and infections may vary based on the clinical manifestations and treatments of individual patients and require careful monitoring of antibody titers post vaccination to identify optimal vaccination schedules in autoimmune individuals.

## 5. Conclusions

This case study exemplifies that hybrid immunity induced by prior SARS-CoV-2 infection and two doses of mRNA vaccination may not be sufficient to provide complete protection from symptomatic Omicron infection in autoimmune patients even without immunosuppressive drugs and requires additional vaccination strategies for this population, taking into account antibody decay post infection and vaccination.

## Figures and Tables

**Figure 1 vaccines-10-00450-f001:**
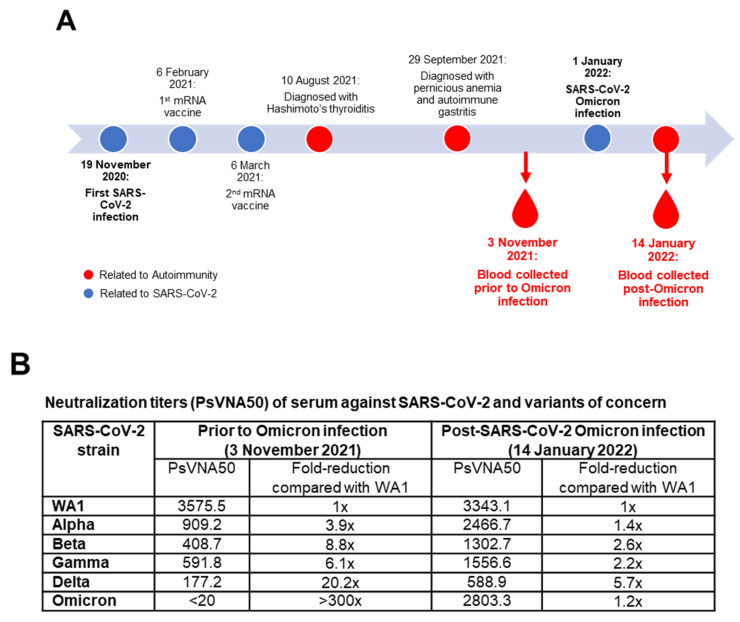
Effect of SARS-CoV-2 vaccination induced immune response and breakthrough Omicron infection in autoimmune patient. (**A**) Timeline of SARS-CoV-2 infections, mRNA vaccinations, and autoimmune diagnosis in relation to Omicron re-infection. (**B**) SARS-CoV-2 neutralizing antibody titers as determined by pseudovirus neutralization assay in 293-ACE2-TMPRSS2 cells with SARS-CoV-2 vaccine homologous WA1/2020 strain, and variants of concern (VOCs): Alpha, Beta, Gamma, Delta, and Omicron in a SARS-CoV-2 PsVNA. PsVNA50 (50% neutralization titer) titer values are shown two months before and 14 days after Omicron infection. Fold reduction (i.e., reduction in neutralization titer) in PsVNA50 titers for each VOC, compared with the prototype WA1/2020 vaccine strain, is shown in the right columns.

## Data Availability

All data needed to evaluate the conclusions in the article are present in the manuscript.

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
