# Peer review of "Immune Response to SARS-CoV-2 Vaccine and Following Breakthrough Omicron Infection in an Autoimmune Patient with Hashimoto’s Thyroiditis, Pernicious Anemia, and Chronic Atrophic Autoimmune Gastritis: A Case Report"

_vaccines, 2022, doi:10.3390/vaccines10030450_

Round 1
Reviewer 1 Report
The manuscript is scientifically sound but has its limitations as a case report. Many comorbidities, individual variability, and SARS-CoV-2 strain, may influence COVID-19 outcome. Efficacy of each available vaccine is variable. A discussion about efficacy of current vaccines should be included, because they have been shown to not be effective in patients with no comorbidities including patients infected with the Omicron variant. Even with these limitations, the manuscript deserves to be published because it provides an evaluation model to guide assessment of other clinical cases. I recommend to further discuss these limitations in the paper prior to publication. In a case report, limitations should not be hidden.
The manuscript is very well written, although some spaces are missing or should be removed.
Author Response
Response:
We thank the reviewer for acknowledging importance of our study. We fixed the minor typos. We have addressed the comments:
Lines 151-159: Prior reports on Omicron breakthrough infections after vaccination with mRNA vaccines did not mention rate of infections among individuals with hybrid immunity and did not report virus neutralizing titers prior to Omicron infection [12-14]. In this case report we found low titers against several VOC and no titers against Omicron prior to second infection. Yet, this patient developed a robust neutralizing antibody titer against Omicron and surprisingly strongly boosted cross-reactive neutralizing antibodies against the other VOCs, while the response against the ancestral prototype spike (Wuhan-like) that was in the two vaccine doses, was not boosted.
Lines 162-164: The limitation of this study is the single case. But the findings warrant further evaluation of breakthrough infections in individuals with hybrid immunity.
Reviewer 2 Report
This case report presents a case of an autoimmune patient about neutralization antibody titers to various strains of SARS-CoV-2 virus after SARS-CoV-2 infection twice and two doses of mRNA vaccine. The report case is a little complicated but seems worth reporting.
The patient was diagnosed with Hashimoto’s thyroiditis, pernicious anemia, and chronic atrophic autoimmune gastritis. The data concerning these diseases such as the level of autoantibodies are available before and after SARS-CoV-2 Omicron infection, they would be interesting.
Author Response
Response:
We thank the reviewer for a positive comment.
Reviewer 3 Report
This is a case study on a single auto-immune patient. The authors have measured neutralizing antibody titres against SARS-COV-2 variants after immunization with the Pfizer mRNA vaccine and a natural infection by the Omicron. The authors observed that a break through infection by the Omicron variant. was still possible despite full immunization and natural infection. However, the patient was sufficiently well protected by the immunization to have escaped hospitalization and death despite his/her auto-immune status. My first concern is that the authors have observed only one patient, and that the results obtained differ from previous similar cases cited by the authors. They should should discuss this difference to enlighten the readership. A second concern is that the patient is one of the co-authors. Generally, self-experimentation is frowned upon. However, since this has been done it should be declared as a potential conflict of interest.
Finally there are many grammatical errors in the text: the text should be thoroughly revised to eliminate these errors.
Author Response
Response:
As per reviewer suggestions, we have added limitation and changed conflict of interest.
Lines 162-164: The limitation of this study is the single case. But the findings warrant further evaluation of breakthrough infections in individuals with hybrid immunity.
Lines 174-175:
Competing Interests: E.C provided plasma samples for evaluation of neutralizing antibodies. All other authors declare that they have no competing interests.